# Randomized trial to compare acceptability of magnesium sulphate administration for preeclampsia and eclampsia: Springfusor pump versus standard of care

Sam Ononge[1]*, Annettee Nakimuli[1], Josaphat Byamugisha[1], Moses Adroma[1], Paul Kiondo[1], Thomas Easterling[2], Hillary Bracken[3]

1 Department of Obstetrics and Gynaecology, Makerere University College of Health Sciences, Kampala, Uganda, 2 University of Washington, Seattle, Washington, United States of America, 3 Gynuity Health Projects, New York, NY, United States of America

* ononge2006@yahoo.com

## Abstract

### Introduction

In low-resource settings, magnesium sulphate ($MgSO_4$) for preeclampsia is administered majorly through an injection into the gluteal muscles 4-hourly for 24 hours. The repeated injections are very painful and may lead to infection, abscess formation, and reduced compliance.

### Objective

To determine the acceptability of Springfusor® pump for the administration of Magnesium Sulphate in preeclampsia and eclampsia.

### Design

Randomized Open Label Clinical Trial.

### Methods

The study was conducted at Kawempe National Referral Hospital. Eligible women had a systolic blood pressure of $\geq$140mmHg and or diastolic blood pressure >90mmHg, proteinuria $\geq$+1, and the physician's decision to start on $MgSO_4$. Four-hundred-ninety-six participants were randomized to a Springfusor® pump group (n = 248) or control (standard of care) (n = 248) administration of $MgSO_4$. Intervention group had a loading dose (4gm of 50% $MgSO_4$ intravenously over 20 minutes) and maintenance therapy (1gm of 50% $MgSO_4$ intravenously per hour for 24 hours) administered using the Springfusor®. The standard of care (SOC) group received a loading dose of 4gm of 20% $MgSO_4$ IV over 15–20 minutes, followed by 10gm of 50% $MgSO_4$ intramuscular (5gm in each buttock) and a maintenance dose of 5gm of 50% $MgSO_4$ was administered IM every 4 hours for 24 hours. Both arms received the rest of the care for preeclampsia/eclampsia as per the hospital guidelines.

**Data Availability Statement:** All relevant data are within the manuscript and its Supporting Information files.

**Funding:** The work was supported by Grant Number D43TW010132 supported by Office Of The Director, National Institutes Of Health (OD), National Institute Of Dental & Craniofacial Research (NIDCR), National Institute Of Neurological Disorders And Stroke (NINDS), National Heart, Lung, And Blood Institute (NHLBI), Fogarty International Center (FIC), National Institute On Minority Health And Health Disparities (NIMHD). Its contents are solely the responsibility of the authors and do not necessarily represent the official views of the supporting offices. The funders had no role in study design, data collection and analysis, decision to publish, or preparation of the manuscript. Go Medical, Subaico, Australia, provided the Springfusor pump and flow-control tubing at a reduced rate but did not contribute to the design or analysis of the study.

**Competing interests:** The authors have declared that no competing interests exist.

Acceptability of the method of administration was assessed using a Likert scale (1–5; 1 and 2: acceptable and 3–5: unacceptable). Pain at the site of $MgSO_4$ administration was assessed using a Visual Analogue Scale 1–7, (1 minimal pain and 7 worst pain). Comparisons were assessed with the Chi-square test, Mann Whitney-Wilcoxon test, and Students' t-test.

## Results

Intervention arm; was more acceptable than the standard of care arm, (95.3% vs70.3%; p<0.001), had a lower median pain score, (2(CI: 2–2), vs 4(CI: 4–5) p<0.001), and fewer side effects. Maternal mortality was comparable between groups (0.8% in the intervention arm vs 1.2% in the IM arm).

## Trial registration

Trial No PACTR201712002887266 (https://pactr.samrc.ac.za/).

## Introduction

Preeclampsia is a multisystem disorder that presents with a raised blood pressure and proteinuria in pregnancy [1]. Globally, preeclampsia complicates approximately 2–8% of the pregnancies [2]. And in Uganda, the incidence of preeclampsia is 4.3% [3]. The presence of convulsions with preeclampsia indicates eclampsia. Preeclampsia and eclampsia (PE/E) are life-threatening for both the mother and the fetus [4], and they are among the leading causes of maternal deaths and disability worldwide, especially in the low-resource setting [5]. The World Health Organization (WHO) estimates that 16% of maternal deaths in low-resource settings are due to PE/E [6]. Magnesium Sulphate ($MgSO_4$) is the drug of choice for prevention and treatment of eclampsia [6]. It is administered parenterally by intravenous (IV) and or intramuscular (IM) routes. The IV therapy is commonly administered following the Zuspan regimen [7] that requires an initial loading dose of 4 gm of magnesium sulphate over 15–20 minutes, followed by 1–2 gm hourly maintenance dose for 24 hours after the loading dose or the last eclamptic seizure. The Zuspan regimen is best delivered by electronic infusion pumps. These electronic pumps are expensive and require electricity or battery to run, making them less appropriate in low-resource settings. In many low-resource settings, $MgSO_4$ administration follows the Pritchard regimen [8]. The regimen is particularly complex and requires both the IV and IM administration of $MgSO_4$. The loading dose of 4 gm is often delivered using an IV push, in which a clinician slowly injects magnesium sulphate with a syringe over 15–20 minutes. This is immediately followed by IM injection of 10 gm of magnesium sulphate into the gluteal muscles (5 gm on each buttock). The maintenance dose of 5 gm IM injection is administered every 4 hours for 24 hours. These repeated IM injections are painful and can increase the risk of abscess development [9]. Because of pain associated with the IM injection, some providers do not administer maintenance therapy and patients too may discontinue the maintenance dose for the same reason. In addition, the Pritchard regimen requires different dilutions for IV and IM doses, and different doses for loading, and maintenance doses. This regimen requires a 20% dilution of magnesium sulphate for the IV loading dose, which necessitates the health providers to calculate the quantity of sterile water to add to the magnesium sulphate solution. In most settings, health providers do not encounter eclampsia very often; and when they do, trying to remember the complex regimen is challenging [10].

In settings with limited resources where electronic infusion pumps are not affordable, there is a need to explore alternative devices that can effectively and safely deliver $MgSO_4$ at a lower cost, while also being acceptable to both the patient and health care provider. The Springfusor® pump and flow control tube (FCT) designed by Go Medical Industries Pty Ltd based in Australia [11], is a promising alternative to the Pritchard method of administering $MgSO_4$, and is designed to simplify continuous IV infusions. The Springfusor pump is an innovative medical device that does not require electricity and it is reusable. It is powered by the potential energy stored within a spring at the heart of the device. The spring is compressed by the action of loading the Springfusor with the compatible syringe and FCT. The spring provides a constant force to the barrel of the loaded syringe. The FCTs are designed to fit easily to the patient's cannula. They come in different flow rates, allowing the user to achieve the require output for precise IV delivery. For our study we utilized two types of FCT, the loading and maintenance dose. While the Springfusor pump can be reused indefinitely on several patients, the FCT must be replaced after each use.

The Springfusor infusion pump is simple to use and setting up, and requires only minimal training to load and operate. It is lightweight, portable, and therefore does not limit the mobility of the patient.

The Springfusor has been used to administer $MgSO_4$ in the treatment of severe preeclampsia in India [12, 13]. In India, Mundle and colleagues compared the manually administered IV loading dose followed by maintenance therapy given by IM route of administration via a syringe, to a loading dose and maintenance therapy given through IV infusion administered by a Springfusor device. Though there were no differences in maternal and neonatal morbidity, the Springfusor had few side effects [13]. Later, Easterling et al. compared the Springfusor administration of continuous IV infusion of magnesium sulphate and 2 hourly IV boluses, the clinical findings were not different in the two groups [12]. Earlier in 1994, Freebairn et al. in Australia compared the Spingfusor infusion device to intermittent boluses on administration of a muscle relaxant. They were able to show that Springfusor provided a more constant level of paralysis compared to intermittent bolus administration [14]. The objective of this study was to demonstrate that using Springfusor pump for intravenous delivery of magnesium sulphate in the treatment of preeclampsia and eclampsia is safer and more acceptable than the standard of care.

## Methods

The trial protocol is submitted as a supplementary file (S2 File. Protocol). The study was a randomized open-label clinical trial conducted at Kawempe National Referral and Teaching Hospital in Kampala district Uganda. The hospital is a government public facility and its Maternal Fetal Unit admits and delivers approximately 2100 pregnant women per month. From the facility records, 7% of the admissions are preeclamptic/eclamptic women. The women with preeclampsia and eclampsia are managed by cadres of health workers ranging from senior consultant obstetricians to medical officers, while the midwives administer the medication and nursing care.

The study included women aged 15 years and above, with a pregnancy of 20+ weeks of gestation or had childbirth within 24 hours, presenting with preeclampsia and eclampsia i.e., have a raised blood pressure (systolic of $\geq 140$ mmHg and/or diastolic $\geq 90$mmHg), proteinuria $\geq 1$ +. We excluded women who had received $MgSO_4$ 24 hours before admission or had known allergy to $MgSO_4$ and known elevated serum creatinine ($>1.2$ mg/dl) before enrolment.

Intervention: Two hundred and forty-eight women randomized to the intervention arm (Springfusor® group) had the loading and maintenance therapy of $MgSO_4$ using IV infusion

administered using a Springfusor® infusion pump. The loading dose was 4 gm of 50% $MgSO_4$ in a 10 ml syringe administered over 20 minutes. The infusion rate was determined by the flow control tubing calibrated to deliver 10 ml of saline over 5 minutes (this system was demonstrated to deliver 4gm of 50% MgSO4 in 20 minutes) [13]. The loading dose was immediately followed by the maintenance dose. The maintenance dose of 4 gm of 50% $MgSO_4$ in 8 ml was administered over 4 hours and the infusion rate was determined by a second flow control tubing calibrated to deliver 10 ml of saline over 60 minutes (this system was demonstrated to deliver 4gm of 50% MgSO4 in 4 hours) [13]. The 4-gm dose of $MgSO_4$ was repeated every 4 hourly for 24 hours. However, if the 4-gm infusion was completed in the less than 4 hours, the next dose was not started not until the 4 hourly interval.

Two hundred forty-eight women randomized to the control arm (standard of care) received $MgSO_4$ administered according to the standard hospital practice (Pritchard regimen). The loading dose of 4 gm of 20% $MgSO_4$ in a 20 ml syringe was administered using an IV infusion over 15–20 minutes. This was immediately followed by an IM injection of 10 gm of 50% $MgSO_4$ mixed with 1 ml of lignocaine into the gluteal muscles (5 gm on each buttock). This was followed by a maintenance dose of 5 gm IM injection, administered every 4 hours for 24 hours.

## Study procedure

Between March and September 2019, women admitted to Kawempe Hospital maternal-fetal unit were screened and consecutively enrolled in the study if they met the inclusion criteria. The study team obtained written informed consent from the participant or the relative if the mother was eclamptic. However, those with eclampsia provided individual written consent later when they regained their consciousness for them to continue participating in the study. Both arms received the care as per national guidelines which included management of high blood pressure using antihypertensives, laboratory investigations (urine analysis, complete blood count, renal function tests, and liver function tests), prevention/treatment of seizures and delivery as planned by the attending physicians. Upon enrolment, study participants were randomized to intervention or standard of care arms. The study nurse assessed the time it took to administer the loading and maintenance doses in both arms using a stop clock. The time was measured from the loading of the $MgSO_4$ into the syringe to the completion of administration of the medicine in the syringe to the patient. For this paper, we report the duration (minutes) it took to administer the loading dose and the second maintenance dose.

The study participants' respiratory rates were monitored by the study midwives every 5 minutes during the loading dose for 30 minutes and hourly during the 24-hour maintenance dose administration. In addition, urine output and tendon reflexes were monitored hourly and documented in the source documents. The side effects of $MgSO_4$ like nausea, vomiting, flushing of the skin, muscle weakness, confusion, and drowsiness were reported 4 hourly and were captured in a questionnaire. The study participants were followed till discharge from the hospital and pregnancy outcomes were extracted from the participants' records or charts.

The primary outcome was acceptability of the method of $MgSO_4$ administration assessed using a Likert scale (1–5; 1: acceptable and 5: unacceptable). The questionnaire was administered after the last maintenance dose. The method of $MgSO_4$ administration was regarded acceptable if the participant rated it 1 or 2, and was unacceptable if the participant scored it 3 or more. Women who discontinued the method of administration by choice or due to side effects were considered in the group of the unacceptable. The secondary outcomes were pain during administration, complications experienced, discontinuation and reliability of the

Springfusor to deliver the $MgSO_4$ as planned. The level of pain experienced during administration of $MgSO_4$ was assessed immediately after the last dose of $MgSO_4$ using a visual analogue scale 1–7 [13], the least (one) representing no pain and the maximum (7) representing the worst pain imaginable. The complications in two arms were measured as the proportion of preeclamptic women who developed any one of the following; $MgSO_4$ toxicity, abnormal liver and renal function test as reported by the laboratory report, an infection at the injection as reported by the clinician or a maternal death. The discontinuation rate was assessed as study participants who did not complete the recommended doses of $MgSO_4$ in 24-hour period. Participants who discontinued the method of administration by choice, due to side effects, or by health worker's decision were counted in each group. And lastly, reliability of Springfusor® pump in the delivery of $MgSO_4$ was measured as the time taken by the device to deliver the $MgSO_4$ as per the clinical recommendations.

## Sample size, randomisation and statistical analysis

The sample size was calculated based on the primary end-point (the proportion of women who reported administration of $MgSO_4$ was acceptable. To calculate the sample size, a two-sided significant level of 5% and power of 90% were chosen. Based on the current literature, the acceptability of $MgSO_4$ administration using the Pritchard regimen (IM maintenance) in the Mundle et al. trial was 31% [13]. To detect an absolute increase of 50 percent points in the proportion of women who reported administration of mgso4 was acceptable between those who used Springfusor and those who used standard of care, we need to have a sample size of 241 in each arm.

Randomization was performed by a biostatistician not involved in the clinical trial who developed the allocation sequence using an online computer random number generator in block sizes of 4 and 6. The allocation sequence was concealed from the research team enrolling and screening participants in serially numbered sealed opaque envelopes (concealed allocation) containing the randomization group. After the research nurse had obtained the consent, she opened the next envelope to determine the group assignment only after the participant was enrolled, completed the baseline assessment and it was time to allocate the intervention. Because of the nature of the study, it was difficult to blind the implementation of the allocation and measurement of the outcome.

Acceptability and safety data were evaluated using intention-to-treat (ITT) analysis. The analysis was conducted using Stata version 12. Descriptive statistics (frequencies, means (standard deviation), median, and ranges) were used to summarize baseline characteristics of study participants and assess if randomization was successful. Student's t-test was performed to compare variables that followed a Gaussian distribution. The Chi-square test was used to evaluate the association between two categorical variables. The Mann Whitney-Wilcoxon test was used to compare the median scores between the two groups. The statistically significant was set at $p < 0.05$.

## Ethical considerations

Approval for this research was provided by the Makerere University School of Medicine Research and Ethics Committee (REC Ref 2018–015) and the Uganda National Council for Science and Technology (HS 2365). Study participants provided written informed consent. The trial was registered with Pan African Clinical Trails PACTR201712002887266. The results are reported in accordance with the CONSORT statement for randomized control trials [15] and the checklist is provided as supporting file (S3 File. Checklist).

## Results

Participants flow: Eligible participants were recruited from March to September 2019. Nine thousand eight hundred twenty-eight (9828) women admitted at the maternal fetal unit of Kawempe National Referral Hospital during the study period were screened for eligibility as shown in consort diagram (Fig 1).

A total of 496 eligible participants were randomized to intervention or standard of care. The loss-to-follow-up was similar in both arms. The 5 patients referred to other facilities were for renal consultations to rule out possible acute kidney injury and Mulago Women

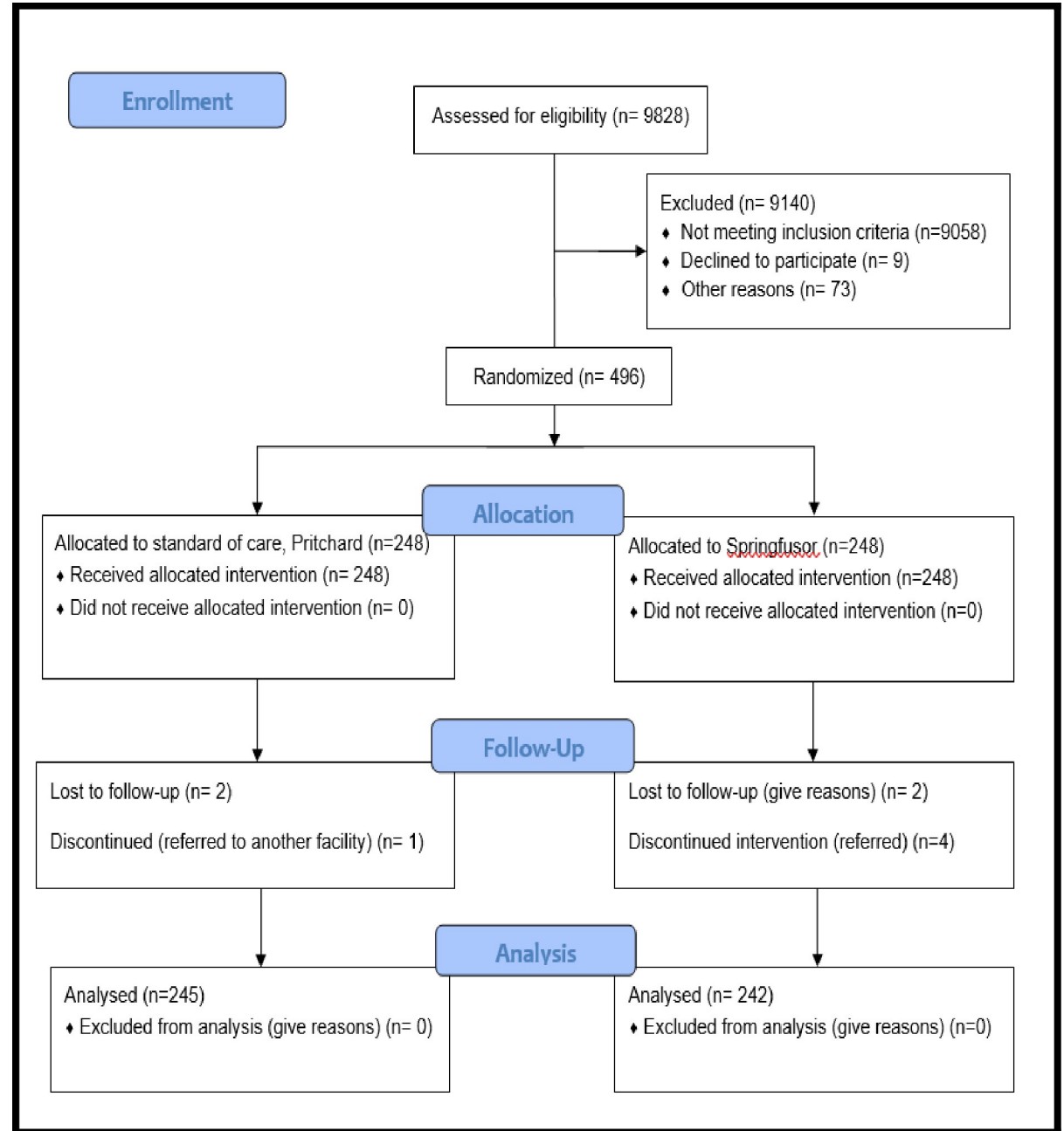

**Fig 1. Consort diagram showing participants flow.**

Specialized Hospital to decongest Kawempe National Referral Hospital when it was overloaded with patients.

## Baseline characteristics

Table 1 shows the baseline characteristics of the women enrolled in the control and interventions arms. The baseline data established that randomization of the two arms were similar for

**Table 1. Demographic and clinical characteristics of the study participants.**

| | Standard of care (Pritchard) arm N = 248 | Springfusor arm N = 248 |
|---|---|---|
| Age in years | | |
| Mean (SD) | 27.2 (5.5) | 26.5 (5.6) |
| Median (Range) | 27 (17–42) | 26 (16–45) |
| Gravidity n (%) | | |
| 1 | 83 (33.5) | 94 (37.9) |
| 2 | 41 (16.7) | 39 (15.7) |
| 3 | 37 (15.1) | 44 (17.7) |
| 4 | 30 (12.1) | 29 (12.0) |
| 5+ | 44 (18.0) | 35 (11.1) |
| Gravida; mean (SD) | 2.7 (1.7) | 2.5 (1.7) |
| Enrolled postpartum (given birth) n (%) | 13 (5.3) | 7 (2.8) |
| Gestation age in weeks at enrollment n (%) | | |
| 20–33 | 74 (29.8) | 61 (24.6) |
| 34–37 | 78 (31.5) | 94 (37.9) |
| 38+ | 79 (31.9) | 79 (31.9) |
| Unknown/missing/postpartum | 17 (6.9) | 14 (5.6) |
| Gestation age in weeks at enrolment; mean (SD) | 34.4 (4.8) | 34.8 (4.6) |
| Gestation age in weeks, median (range) | 36 (20–43) | 36 (20–43) |
| Number of ANC visits (%) | | |
| None | 10 (4.0) | 8 (3.2) |
| 1–3 | 145 (58.5) | 142 (57.3) |
| 4+ | 93 (37.5) | 98 (39.5) |
| Number of ANC visits; mean (SD) | 3.0 (1.3) | 3.2 (1.3) |
| Multiple pregnancy (%) | 21 (8.5) | 14 (5.8) |
| Admitted as n (%) | | |
| Patients referred in | 179 (72.2) | 169 (70.1) |
| Self-referrals in (Walk-in) | 21 (8.5) | 34 (13.7) |
| Antenatal clinic at study site | 48 (19.5) | 38 (15.8) |
| Systolic BP at enrolment mmHg | | |
| Mean (SD) | 168.8 (22.4) | 167.0 (21.0) |
| Median (Range) | 163 (126–237) | 166 (130–270) |
| ≥160 mmHg (%) | 145 (58.5) | 151 (60.9) |
| Diastolic BP at enrolment mmHg | | |
| Mean | 111.5 (14.6) | 112.3 (13.2) |
| Median (Range) | 110 (78–173) | 112 (90–162) |
| ≥110 mmHg (%) | 129 (52.0) | 142 (57.3) |
| Enrolled as Eclamptic (%) | 14 (5.6) | 11 (4.4) |

ANC = antenatal care, BP = blood pressure, SD = standard deviation

**Table 2. Laboratory characteristics of the participants enrolled into the study.**

|  | Standard of care N = 248 (%) | Springfusor arm N = 248 (%) |
|---|---|---|
| Low platelets ($<100$x$10^3$/µmoll) | 20 (8.1) | 29 (11.7) |
| Elevated alanine transaminase ($>31$µmol/L) | 46 (18.5) | 40 (16.1) |
| Elevated aspartate transaminase ($>32$µmol/L) | 70 (28.2) | 69 (27.8) |
| Raised bilirubin ($>3.4$µmol/L) | 87 (35.1) | 80 (32.2) |
| Elevated serum creatinine ($\geq1.2$mg/dL)* | 20 (8.1) | 38 (15.3) |

*Almost all the results were available for the attending physicians after the prevention and treatment of seizures with magnesium sulphate was initiated or completed. None of the participants with elevated serum creatinine was excluded from the study.

almost all variables except for those presenting as a postpartum and multiple pregnancy. More than two-thirds of participants were patients referred from lower facilities.

Table 2 shows the laboratory parameters of the study participants enrolled in the control and intervention arms. The baseline laboratory parameters were similar in the two groups with the exception for elevated serum creatinine in the intervention arm.

As shown in Table 3, almost all women in the intervention group found Springfusor administration of $MgSO_4$ acceptable (95.3%, CI: 91.2–97.1) compared to the standard of care (70.3%, CI: 64.3–75.9) and p$<$0.001. The women in the standard of care arm had a higher median pain score than the intervention arm (standard of care: 4(IQR: 4–5); Springfusor: 2(IQR: 2–2), p$<$0.001). More women in the intervention arm would recommend Springfusor method of administration of $MgSO_4$ to a friend compared to standard of care (96.2% vs 61.4%, p$<$0.001). Similarly, almost all women in the intervention arm would use the Springfusor for magnesium administration when they have raised blood pressure in the next pregnancy compared to the control arm (Pritchard method) (95.7% vs 65.7%, p$<$0.001).

The duration of $MgSO_4$ administration of loading dose was longer in the standard of care than in Springfusor arm (Table 4). Few women in both groups discontinued $MgSO_4$ (did not complete the recommended 24-hour doses) and the rates were similar (5.3% vs 5.0% p = 0.862).

**Table 3. Acceptability of $MgSO_4$ administration using Springfusor compared to standard of care.**

|  | Standard of care arm (Control) N = 236 | Springfusor arm N = 235 | p-value |
|---|---|---|---|
| **Primary outcome** |  |  |  |
| Acceptable x$\leq$2 score on Likert scale n(%, CI) | 166 (70.3, CI: 64.3–75.9) | 224 (95.3, CI: 91.2–97.1) | $<$0.001* |
| **Secondary outcomes** |  |  |  |
| Discontinuation; Not completed the 24 hours n (%, CI) | 13 (5.5, CI: 3.4–9.4) | 12 (5.1, CI: 2.5–8.1)) | 0.862* |
| Median pain score during administration (median IQR) | 4 (IQR: 4–5)) | 2 (IQR: 2–2) | $<$0.001+ |
| Would recommend method of administration of $MgSO_4$ to friend n(%, CI). | 145 (61.4, CI: 54.8–67.2) | 226 (96.2, CI: 93.3–98.3) | $<$0.001* |
| Would consider using it again in future if she experienced a raised BP during her next pregnancy n (%, CI) | 155 (65.7, CI: 59.6–71.8) | 225 (95.7, CI:93.2–98.3) | $<$0.001* |

*Chi Square test and +Mann Whitney-Wilcoxon test for the pain score, CI = confidence interval, IQR = interquartile range

**Table 4. Drug administration and pain score registered by study participants.**

|  | Standard of care (control) arm | Springfusor arm | p value |
|---|---|---|---|
| Duration in minutes of loading dose magnesium administration; mean (SD) | 25.8 (8.3) | 21.1 (5.8) | <0.001[+] |
| Duration of administering the 2nd Maintenance (2nd 4 hours) in minutes, mean (SD) | 2.8 (3.5) | 236.3 (40.2) | N/A* |
| Discontinued MgSO4 administration before 24 hours n (%) | 13 (5.3) | 12 (5.0) | 0.862[++] |
| Reasons for discontinuation of MgSO$_4$, n (%) |  |  |  |
| Doctor's advice | 7 (53.8) | 7 (58.3) | 0.863[++] |
| Patients request | 4 (30.8) | 4 (33.3) |  |
| Referred to another facility before completion | 2 (15.4) | 1 (8.3) |  |
| Pain score after administration of MgSO$_4$: 1(minimal), 7 (worst) n(%) |  |  |  |
| 1 | 11 (4.5) | 71 (29.3) | <0.001[++] |
| 2 | 31 (12.6) | 97 (40.1) |  |
| 3 | 48 (19.5) | 38 (15.7) |  |
| 4 | 42 (17.1) | 6 (2.5) |  |
| 5 | 55 (22.4) | 14 (5.8) |  |
| 6 | 28 (11.4) | 5 (2.1) |  |
| 7 | 21 (8.5) | 4 (1.7) |  |
| missing | 10 (4.1) | 7 (2.9) |  |

SD = standard deviation. [+]student t-test, and [++]Chi-sqaure test,.

*Not provided statistic, because the Springfusor system is designed to deliver the drug in four hours and comparing it with intramuscular administration which takes two minutes or less. Using statistics to compare this will be misinterpreted and provides no meaning.

The side effects like flushes, nausea, vomiting, drowsiness, and diplopia were more in the standard of care than in Springfusor (Table 5). However, the adverse events like respiratory depression, depressed patellar reflex and cardiac arrest were very few and were comparable in the two arms.

Approximately 6% of the participants were discharged from the hospital while still pregnant after the blood pressures were controlled (Table 6). Though outcomes of pregnancy were comparable in both arms, more women were delivered by caesarean section in the Springfusor arm than in the SOC. There were approximately 3% of the participants were referred to renal physicians for specialized care (including dialysis) and follow-up. Unfortunately, there were 5 maternal deaths in the whole study.

**Table 5. Adverse and side effects experienced during the 24 hours of drug administration.**

|  | Total N = 487(%) | Standard of care (control) arm n = 245 (%) | Springfusor arm n = 242 (%) |
|---|---|---|---|
| Flushes | 373 (76.8) | 201 (80.0) | 172 (71.4) |
| Nausea | 120 (24.7) | 70 (28.6) | 50 (20.8) |
| Vomiting | 56 (11.5) | 33 (13.5) | 23 (9.5) |
| Headache | 107 (22.0) | 52 (21.2) | 55 (22.8) |
| Drowsiness | 164 (33.7) | 97 (39.6) | 67 (27.8) |
| Diplopia | 16 (3.3) | 7 (2.9) | 9 (3.7) |
| Burning/warm sensation at the site of injection | 136 (28.0) | 68 (27.8) | 68 (28.2) |
| Respiratory depression <16 breaths/min | 2 (0.5) | 1 (0.5) | 1 (0.5) |
| Depressed patellar reflexes | 1 (0.5) | 1 (0.5) | 0 (0.0) |
| Cardiac arrest | 2 (0.5) | 1 (0.5) | 1 (0.5) |

**Table 6. Maternal and newborn outcomes of the study participants.**

| | Total (%) | Standard of care (control) arm (%) | Springfusor arm (%) |
|---|---|---|---|
| Discharged while still pregnant | 28 (5.8) | 18 (7.4) | 10 (4.1) |
| Mode of delivery (n = 458) | | | |
| Caesarean | 201 (43.9) | 80 (35.2) | 121 (52.4) |
| Vaginal | 257 (56.1) | 147 (64.8) | 110 (47.6) |
| Outcome of the delivery (fetus)n = 488 | | | |
| Alive | 398 (81.6) | 195 (80.9) | 203 (82.5) |
| Still births | 84 (17.2) | 43 (17.8) | 41 (16.7) |
| missing | 6 (1.2) | 3 (1.2) | 3 (1.2) |
| Weight of the baby | | | |
| <2500 gm | 244 (50.5) | 126 (52.3) | 118 (48.8) |
| ≥2500 gm | 239 (49.5) | 115 (47.7) | 124 (51.2) |
| Referred to Nephrology hospital for further renal care n = 487 | 14 (2.9) | 8 (3.3) | 6 (2.5) |
| Maternal death | 5 (1.0) | 3 (1.2) | 2 (0.8) |
| No convulsion after loading dose | 0 | 0 | 0 |

## Discussion

The definitive treatment of preeclampsia is the delivery of the fetus and placenta. However, before and after delivery of the baby and placenta, the goal of management is to control the blood pressure to normal range and minimize the development of complications like

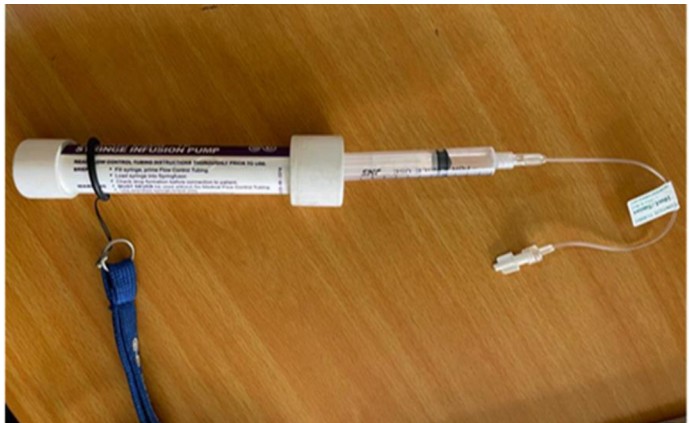

Springfusor$^R$ pump with 10ml syringe and flow control tubing

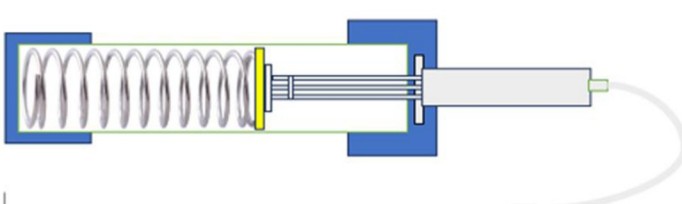

Illustration of assembled Springfusor$^R$ pump with syringe and FCT

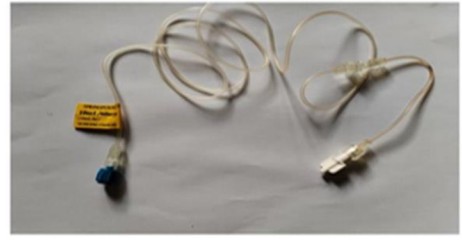

Flow Control Tubing for loading dose administration

Flow control tubing for maintenance dose administration

**Fig 2. The Springfusor pump with a 10 ml syringe, and flow control tubing (FCT).** The Springfusor pump applies constant pressure to a syringe using a simple spring mechanism. The infusion rate is determined by a specific FCT, which provides consistent resistance to produce a steady flow. The FCT for the loading dose delivers 4 grams of 50% MgSO4 over 20 minutes, while the one for maintenance dose administration delivers 4 grams of 50% MgSO4 over 4 hours.

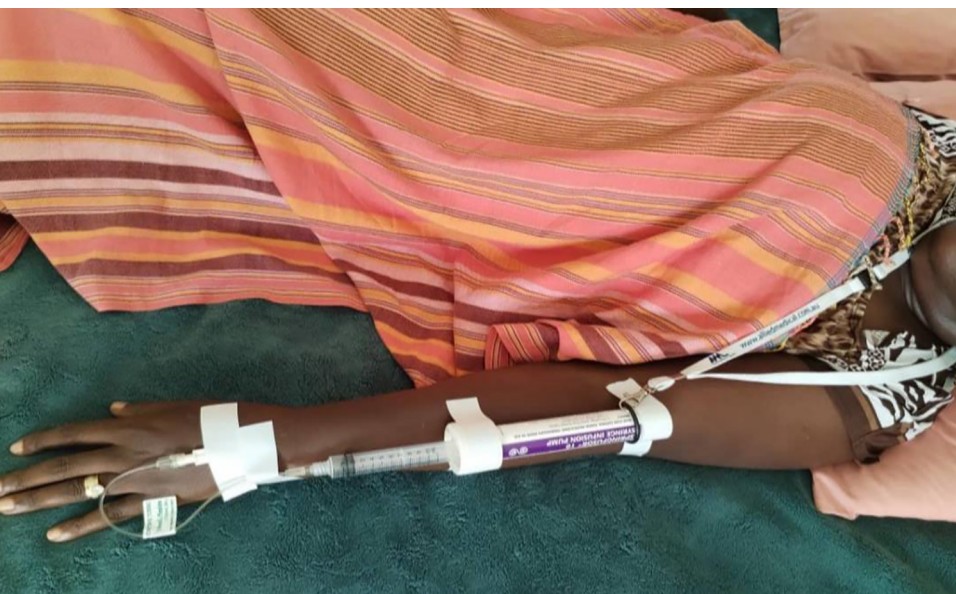

**Fig 3. Showing participant with Springfusor pump with magnesium sulphate administered through a cannula.**
The neck suspension string on the patient's neck enables participants mobility without fear of dislodging the cannula.

eclampsia. $MgSO_4$ is the anticonvulsant for eclampsia prophylaxis and treatment [16]. $MgSO_4$ therapy could be given by continuous IV infusion (Zuspan) [7] or by administering an IV bolus and IM doses for the loading dose followed by IM injections every 4 hours (Pritchard) [8]. In low-resource settings due to unavailability of the electronic pumps, the Pritchard regimen is the standard of care. The loading dose of 4gm of $MgSO_4$ is often delivered via an IV push over 15–20 minutes. This process is challenging for the provider and difficult in a busy ward, and is associated with inconsistent flow rates. If administered faster than the recommended 15–20 minutes, the IV push may lead to increased pain, nausea, vomiting, and flushing. Meanwhile, the repeated IM injections that follow the loading dose are associated with pain, hot flushes, somnolence, and sometimes abscess formation at the site of injection.

We assessed the acceptability and safety of Springfusor pump for intravenous delivery of $MgSO_4$ for prophylaxis and treatment of eclampsia among women admitted at Kawempe National Referral Hospital. Acceptability of intravenous administration of $MgSO_4$ using Springfusor was higher compared to the Pritchard regimen (standard of care). The level of acceptability of intravenous administration of $MgSO_4$ using Springfusor was comparable to a study by Mundle et al in India that found that it was 97% [13]. The low acceptability associated with standard of care is most likely due to pain that follows IM injection. Literature shows that the anxiety and fear associated with pain following IM injection reduces the acceptability of treatment to the patients [17, 18]. With the four-hourly frequency of IM injection of $MgSO_4$ for prophylaxis and treatment of preeclampsia and eclampsia in standard of care, injection site pain is an important concern and local guidelines propose addition of a local anesthetic agent (Lignocaine) into the drug. Despite the addition of 1 ml of lignocaine into every IM injection of $MgSO_4$, our study findings showed that participants on the standard of care experienced higher pain scores than women who received the drug intravenous, administered by the Springfusor pump.

The majority (96%) of the participants in the intervention arm responded that they would recommend IV magnesium sulfate administered using Springfusor to other patients compared to standard of care (61%). In addition, almost all participants in intervention arm (97%)

compared to two-thirds (66%) in standard of care would use $MgSO_4$ in future pregnancy if they get preeclampsia. Literature reports that, when the safety and efficacy of two injection routes are equivalent, health care providers should consider more about patient's preference because it will ensure optimal treatment adherence and ultimately improve patient's experience or satisfaction [19, 20]. In our study, findings showed that no women developed a seizure after enrolment, indicating that both routes and dosing prevented and controlled the convulsions adequately.

Overall, the discontinuation rate was low (5.1%) and there was no difference in the two arms. The low rates of discontinuation could be because this assessment was among participants who participated in a research study environment which has ample opportunities for questions, comments, and explanations of the process. In the real world, it is unlikely that this level of support will be available. Slightly more than half (56%) of the discontinuation was due to physician on duty recommendation. The other reasons for discontinuation were due to participant's requests (32%), and three patients (12%) were referred to seek treatment in another facility before completion of the drug administration.

The Springfusor® pump and flow control tube (FCT) is an encouraging alternative to repeated IM administration of $MgSO_4$, designed to make simpler, the continuous IV infusions. It does not require electricity and it is reusable. The Springfusor is powered by the potential energy stored within a spring at the heart of the device (Fig 2), suitable for low-resource settings.

The FCTs exist in a variety of flow rates which enables the user to attain the desired output for exact IV delivery needs. For this study we used two varieties of FCTs as shown in Fig 2; the loading dose and maintenance dose. The Springfusor syringe infusion pump is a low-cost technology that requires only minimal training to load and operate. Being lightweight, and with a neck strap, it does not limit the mobility of the patient (Fig 3).

This study had some limitations. Firstly, the delay in accessing the laboratory work of the participants resulted in some participants who should be excluded based on serum creatinine, enrolled in the study. Due to the morbidity and mortality associated with PE/E, the standard of care at the facility does not wait for laboratory results before starting $MgSO_4$ prophylaxis. We operated within the hospital guidelines. Fortunately, no participant experienced $MgSO_4$ toxicity. Secondly, we could not blind the study to the participants and providers, and might have influenced the reporting of the outcomes. The nature of the study could not enable blinding of the study.

## Conclusion

Acceptance of prescribed therapy is key for adherence and to clinical outcomes, and the effect is particularly critical for treatments that require repeated injections like $MgSO_4$ for preeclampsia and eclampsia. Pain associated with intramuscular injection (standard of care) was less with the intravenous infusion (Springfusor®) than with intramuscular administration. In addition, the intravenous administration was preferred to the standard of care with women endorsing a greater likelihood to use it in the next pregnancy or recommend it to a friend.

## Supporting information

**S1 File. Springfusor study data.**
(XLS)

**S2 File. Protocol Springfusor for preeclampsia study.**
(DOCX)

**S3 File. CONSORT checklist.**
(DOCX)

## Acknowledgments

The authors are grateful to Kawempe National Referral Hospital administration and staff for support in study implementation, data collectors for the great work done and the participants who gave in their time to participate in the study.

## Author Contributions

**Conceptualization:** Sam Ononge, Annettee Nakimuli, Josaphat Byamugisha, Moses Adroma, Thomas Easterling, Hillary Bracken.

**Data curation:** Sam Ononge, Paul Kiondo.

**Formal analysis:** Sam Ononge, Moses Adroma.

**Funding acquisition:** Sam Ononge, Annettee Nakimuli, Josaphat Byamugisha.

**Methodology:** Sam Ononge, Annettee Nakimuli, Josaphat Byamugisha, Moses Adroma, Paul Kiondo, Thomas Easterling, Hillary Bracken.

**Project administration:** Sam Ononge, Josaphat Byamugisha, Moses Adroma, Paul Kiondo.

**Supervision:** Sam Ononge, Annettee Nakimuli, Josaphat Byamugisha, Moses Adroma, Paul Kiondo, Thomas Easterling, Hillary Bracken.

**Writing – original draft:** Sam Ononge, Thomas Easterling.

**Writing – review & editing:** Sam Ononge, Annettee Nakimuli, Josaphat Byamugisha, Moses Adroma, Paul Kiondo, Thomas Easterling, Hillary Bracken.

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
