## [Decision Letter · Decision Letter 0]

20 Nov 2023

PONE-D-23-13036Randomized trial to compare acceptability of Magnesium Sulphate administration for preeclampsia and eclampsia: Springfusor pump versus standard of carePLOS ONE

Dear Dr. Ononge,

Thank you for submitting your manuscript to PLOS ONE. After careful consideration, we feel that it has merit but does not fully meet PLOS ONE’s publication criteria as it currently stands. Therefore, we invite you to submit a revised version of the manuscript that addresses the points raised during the review process.

**ACADEMIC EDITOR:**

Please respond to all reviewers comments one by one 

We look forward to receiving your revised manuscript.

Kind regards,

Ahmed Mohamed Maged, MD

Academic Editor

PLOS ONE

Journal Requirements:

Reviewers' comments:

Reviewer's Responses to Questions

**Comments to the Author**

1. Is the manuscript technically sound, and do the data support the conclusions?

Reviewer #1: No

Reviewer #2: Yes

Reviewer #3: Partly

2. Has the statistical analysis been performed appropriately and rigorously? 

Reviewer #1: No

Reviewer #2: Yes

Reviewer #3: No

3. Have the authors made all data underlying the findings in their manuscript fully available?

Reviewer #1: Yes

Reviewer #2: Yes

Reviewer #3: Yes

4. Is the manuscript presented in an intelligible fashion and written in standard English?

Reviewer #1: No

Reviewer #2: Yes

Reviewer #3: No

5. Review Comments to the Author

Reviewer #1: A two-arm randomized controlled clinical trial was conducted which aimed to determine the acceptability of Springfusor pump for the administration of Magnesium Sulphate in patients with preeclampsia or eclampsia. The intervention arm had a statistically significant higher acceptance rate than the control arm. The other conclusions are unclear.

Major revision:

The statistical quality of this manuscript is low. Detailed statistical methods are missing, and statistical tests for comparing factors/outcomes between the two groups have been omitted.

Specific revisions:

1- Line 183: Indicate the statistical method which attains 90% power.

2- Line 198: Specify the descriptive statistics used.

3- Line 205: Clarify that the acceptability rates were compared between the two groups using a chi-square test.

4- Line 207: The level of pain may be summarized as means and standard deviations, but means is not a statistical testing method for conducting comparisons. Typically pain scores are not normally distributed. Test the distribution of pain scores for normality. If the data is normally distributed or can be transformed to a normal distribution then summarize with the mean and standard deviation and compare groups using a parametric test, such as a t-test. If the distribution is not normal, summarize with median, first and third quartiles and compare using a nonparametric test, such as a Wilcoxon rank sum test.

5- Table 1: When providing ranges also provide medians. Provide p-values for comparing the demographic and clinical characteristics between the two arms. These p-values will support concluding statements in lines 226-230. Include the p-values in the text as well.

6- Lines 234-5: Provide p-values for comparing platelets counts and creatinine levels in the two arms.

7- Table 2: Add p-values for comparing the two arms.

8- The standard statistical term for average is mean.

9- Line 244: Provide further clarity about why the p-value is not provided (NA). Provide p-values for comparing the categorical factors in table 3. Due to the small sample sizes, Fisher’s exact tests may be more powerful than chi-square tests.

10- Line 247: The word “result” is missing from “a statistically significant result”.

11- Line 251: Provide a p-value for comparing 96% to 61% or provide 95% confidence intervals.

12- Table 5: Provide p-values for comparing the adverse events/side effects in the two arms.

13- Table 6: Provide p-value for comparing pregnancy outcomes in the two arms.

14- Thoroughly proofread the manuscript. There are several grammatical errors.

Reviewer #2: Introduction

Please reference Zuspan and Pritchard methods

Lines 69-85: no reference is made to the literature

Where under 18 years are allowed to consent independently

Please describe the 24 hours mgso4 use, as it is different in PE/E

Define PE/E as an abbreviation

154: fits, perhaps use seizures

177: maternal death

231: table 1 qualify multiple pregnancy*

265: define SOC earlier in the manuscript

272-283 is a repeat from the literature

313: 4 patients declined mgso4, 3 went for a second opinion to another facility, what happened to 1?

Reviewer #3: The authors are to be commended for their research efforts; however, there are some concerns that I want to address about the manuscript:

Abstract:

Please mention how many patients there are in each group.

Introduction:

The authors are encouraged to mention the incidence of preeclampsia in Uganda to indicate the magnitude of the problem related to their context.

Page 4, line 64: “are due to PE/E” When the abbreviation is mentioned for the first time, the extension should be written.

Page 4, line 68; “over 15-20 minutes-over 15-20 minutes (mins)” The phrase has been repeated twice; please correct it.

Many paragraphs are missing references, particularly for the Zuspan regimen and the Pritchard regimen.

Please summarize the introduction.

Methods:

Please mention how many patients there are in each group.

The authors are encouraged to justify why they used a visual analog score from 1 to 7, as the VAS consists of a 10cm line with two end points representing 0 ('no pain') and 10 ('pain as bad as it could possibly be') and asking the patient to rate their current level of pain by placing a mark on the line.

When was the VAS score assessed? The authors mentioned during administration that for the IV, sure, it will be once, but for the IM, it will be more than once. It is logical to have more pain among the standard of care group. So, do the scores taken reflect the first time of administration? It needs to be clarified how the VAS scores were calculated or recorded. If the VAS score was high (>4), what was the management?

Statistical analysis:

The statistical tests were not appropriately described, and the level of significance was not mentioned.

Results:

The authors mentioned in the methodology section that they will exclude patients with creatinine (>1.2 mg/dL); however, they added them to the study as discussed in the results section, which is somehow confusing and declares some safety issues.

The authors included patients with low platelets in the standard of care group, as mentioned in Table 2. Low platelets are considered a contraindication for IM injection for fear of a huge hematoma.

For tables in general, add an explanation to the footnotes of the tables that include “Data presented as...", Types of tests used as labeled in the table, and abbreviations in the table.

Kindly indicate the significance with an asterisk “*” in the table and add the explanation to the footnotes of the tables.

Figure 3: Check the copyright for the figures used.

Grammar:

Kindly revise the whole paper, as there are some grammar mistakes.

Thanks

6. PLOS authors have the option to publish the peer review history of their article (what does this mean?). If published, this will include your full peer review and any attached files.

Reviewer #1: No

Reviewer #2: **Yes: **Shastra Avendra Bhoora

Reviewer #3: No

---

## [Decision Letter · Decision Letter 1]

26 Jan 2024

PONE-D-23-13036R1Randomized trial to compare acceptability of Magnesium Sulphate administration for preeclampsia and eclampsia: Springfusor pump versus standard of carePLOS ONE

Dear Dr. Ononge,

Thank you for submitting your manuscript to PLOS ONE. After careful consideration, we feel that it has merit but does not fully meet PLOS ONE’s publication criteria as it currently stands. Therefore, we invite you to submit a revised version of the manuscript that addresses the points raised during the review process.

**ACADEMIC EDITOR: **

Please respond to all reviewers comments

We look forward to receiving your revised manuscript.

Kind regards,

Ahmed Mohamed Maged, MD

Academic Editor

PLOS ONE

Reviewers' comments:

Reviewer's Responses to Questions

**Comments to the Author**

1. If the authors have adequately addressed your comments raised in a previous round of review and you feel that this manuscript is now acceptable for publication, you may indicate that here to bypass the “Comments to the Author” section, enter your conflict of interest statement in the “Confidential to Editor” section, and submit your "Accept" recommendation.

Reviewer #1: (No Response)

Reviewer #4: (No Response)

Reviewer #5: All comments have been addressed

2. Is the manuscript technically sound, and do the data support the conclusions?

Reviewer #1: Yes

Reviewer #4: Partly

Reviewer #5: Yes

3. Has the statistical analysis been performed appropriately and rigorously? 

Reviewer #1: Yes

Reviewer #4: No

Reviewer #5: Yes

4. Have the authors made all data underlying the findings in their manuscript fully available?

Reviewer #1: Yes

Reviewer #4: Yes

Reviewer #5: Yes

5. Is the manuscript presented in an intelligible fashion and written in standard English?

Reviewer #1: Yes

Reviewer #4: No

Reviewer #5: Yes

6. Review Comments to the Author

Reviewer #1: Minor Revisions:

1- Sample Size justification: The statistical method which attains 90% power has not been stated.

2- For consistency, abbreviate standard deviation SD.

Reviewer #4: Authors performed a randomized trial to compare acceptability of Magnesium Sulphate administration for preclampsia and eclampsia between Springfusor pump versus standard of care in a teaching hospital in Kampala (Uganda). I have several major comments.

1.- Line 45, Authors used an scale to assess the acceptability of both methods assuming a cutoff of 3 for unacceptable, Did you perform any previous analysis to validate that scale and which was your criteria for selecting 3 as the optimal cutoff?

2.- Line 116, The objective is not clear, “to assess the acceptability and safety of Springfusor pump..” but how? Maybe not in here but in the rest of the text you should explain if you are considering a fixed threshold as a definition of acceptable or unacceptable for the intervention, regardless of control, or if you want to demonstrate that the intervention is superior to standard of care, and in this case, you should fix the threshold for clinical significance, see my comments number 4th and 18th.

3.- Line 127, Explain the criteria used to include participants, (consecutive?)

4.- Line 189, Please rewrite the sample size justification, explain that you are going to use a test for the comparison of two proportions, that the expected acceptance proportions are 31% and XXX(see comments 2nd and 18th)

5.- line 193, Explain the reason for considering an expected acceptance proportion of 46.5% in the Springfusor pump. Based on reference [13] this proportion should be higher and therefore it is not clear why you have included so many participants in this study

6.- Line 212, please change this expression: ”means ± standard deviation” by “means (standard deviation)”, it seems as if you were presenting the CI and it is not true.

7.- Line 213, rewrite this sentence “Student´s t test was performed to compare variables in a Gaussian distribution”

8.- Line 214, Change “to evaluate categorical…” by “to evaluate association between two categorical..”

9.- Line 215, rewrite this sentence “The Wilcoxon Rank Sum test was used to evaluate the differences in a non-Gaussian distribution in the two groups.”

10.- Tables 1 and 2, As it is established in the CONSORT guide (Moher et al, 2010) Tests of baseline differences are not necessarily wrong, just illogical. It makes no sense to determine if the difference between groups is due to randomization, we already know it. When using statistical tests, you are wondering if the observed baseline distribution could be extrapolated to different samples, but you are not going to use a different sample, so please remove p-value in these tables.

Moher D, Hopewell S, Schulz KF, Montori V, Gøtzsche PC, Devereaux PJ, Elbourne D, Egger M, Altman DG. CONSORT 2010 explanation and elaboration: updated guidelines for reporting parallel group randomised trials. BMJ. 2010 Mar 23;340:c869. doi: 10.1136/bmj.c869.

11.-Lines 237 and 243, remove statistically significant

12.- Line 244, Table 2, proportions of women with elevated serum creatinine are quite different between both arms, explain if that characteristic may introduce bias in the results or not.   

13.- Provide a table with the proportion of missing data

14.- Page 15, Please order the results, the main objective should be the first one, this is to say, to place Table 4 before Table 3.

15.Line 258, Change “(chi 49.7 p<0.001)” by (effect size for the difference and CI), neither the chi square test value nor the p-value can be considered as an effect size.

16.-Table 4, Please also include the CI for the difference in the first row

17.-Table 4, if you planned to perform an ITT analysis, explain the reasons for just considering N=236 participants per arm in the primary outcome, and how you have dealt with the missing data, I suggest performing a sensitivity analysis to assess the impact of missing data.

18.- The main issue in this study is the initial hypothesis:” To demonstrate an increase of 50% in the intervention arm” and you have not proven it. Acceptable percentage in standard care 70.3% and acceptable percentage in intervention 95.3%, which corresponds to an increase of 35%, do you consider this increment as clinically relevant? If your answer is affirmative, why do you justify the same size assuming a higher increment?

Minor comments

1.- Revise some grammatical mistakes

2.- Line 153, check dates: March and September 2019 or March and August 2019 (line 237)

3.-Lines 205-209, this paragraph is already included in lines 172-175

4.- Improve the presentation in Table 3, sometimes is means (SD) which is included, but others is N(%), others N…try to keep the same format z

5.- Line 252, the footnote is useless "Chi-sqaure test, fishers exact test, student t test and wilxon Rank sum test", explain row by row which one you have used.

6. Line 252, change fishers by Fisher¨s exact test

7.- Line 252, change Wilxon by Mann Whitney-Wilcoxon, to avoid confusion

8.- Table 4, X/Z column is not needed

9.- In the whole manuscript, change Wilcoxon Rank Sum Test by Mann Whitney-Wilcoxon test. Although it is not wrong, the test is commonly named as Wilcoxon when considering two paired samples and Mann Whitney for two independent samples.

10.- Line 260, Z is not needed

11.- Line 266, As in comment 5th, this footnote is useless, explain row by row which one you have used.

12.- Table 6 is a mixture of characteristics, consider changing the name of the title

13.- Improve the presentation of Table 6, use the same names for the columns, "intervention" is not used in previous tables, close the brackets...

Reviewer #5: This paper addresses a critical issue and adds critical knowledge in the field of Obstetrics. Preeclampsia and eclampsia remain important contributors to mortality and morbidity in Uganda and beyond. An alternative way for administration of MgSO4 is expected to improved outcomes, clinician and patient experiences is welcome in the literature. The results clearly show that the new method of administration was acceptable. I recommend the paper is accepted with minor revisions

Comments:

Line 227: .... Approximately, ten thousand (9828) women admitted at the... Remove the redundant first part of the sentence. Mention the exact number of admissions since you have it.

Line 330: ... 5 referrals... please mention 5 patients were referred... if that's what you mean. Also harmonise "referrals" in Table one and else where to mean referred patients

Line 251: Table 3 ... Duration in mins of loading dose magnesium administration; mean(sd). Write mins in full to mean minutes. Also correct hrs if it means hours in line 256 and else where

Generally your table need to be improved

The methods and analyses are well presented. The results are well presented and well discussed.

7. PLOS authors have the option to publish the peer review history of their article (what does this mean?). If published, this will include your full peer review and any attached files.

Reviewer #1: No

Reviewer #4: **Yes: **Teresa Pérez

Reviewer #5: No

---

## [Decision Letter · Decision Letter 2]

2 Apr 2024

PONE-D-23-13036R2Randomized trial to compare acceptability of Magnesium Sulphate administration for preeclampsia and eclampsia: Springfusor pump versus standard of carePLOS ONE

Dear Dr. Ononge,

Thank you for submitting your manuscript to PLOS ONE. After careful consideration, we feel that it has merit but does not fully meet PLOS ONE’s publication criteria as it currently stands. Therefore, we invite you to submit a revised version of the manuscript that addresses the points raised during the review process.

**ACADEMIC EDITOR: Please respond to all reviewers comments**

We look forward to receiving your revised manuscript.

Kind regards,

Ahmed Mohamed Maged, MD

Academic Editor

PLOS ONE

Reviewers' comments:

Reviewer's Responses to Questions

**Comments to the Author**

1. If the authors have adequately addressed your comments raised in a previous round of review and you feel that this manuscript is now acceptable for publication, you may indicate that here to bypass the “Comments to the Author” section, enter your conflict of interest statement in the “Confidential to Editor” section, and submit your "Accept" recommendation.

Reviewer #1: All comments have been addressed

Reviewer #4: (No Response)

2. Is the manuscript technically sound, and do the data support the conclusions?

Reviewer #1: (No Response)

Reviewer #4: Partly

3. Has the statistical analysis been performed appropriately and rigorously? 

Reviewer #1: (No Response)

Reviewer #4: No

4. Have the authors made all data underlying the findings in their manuscript fully available?

Reviewer #1: (No Response)

Reviewer #4: Yes

5. Is the manuscript presented in an intelligible fashion and written in standard English?

Reviewer #1: (No Response)

Reviewer #4: No

6. Review Comments to the Author

Reviewer #1: (No Response)

Reviewer #4: Thanks for addressing the comments. A few revisions are still needed.

Some of them are related to my previous comments:

1.- Line 50, Delete this sentence, “baseline characteristics were similar in both arms”, this is an expected result in randomized trials that always needs to be checking, it is not specific of your study.

2.- Line 114, I think the authors misunderstood my previous comment regarding the objective. I was not asking about the definition of acceptability I am asking about the main aim. Do you want to estimate the percentage of women that consider Springfusor as acceptable? Or do you want to demonstrate the superiority of Springfusor, in terms of acceptability, compared to standard of care? In my opinion your objective is the later, therefore you should change it.

3.- Line 192, delete “to minimize the risk of type II error, you are fixing it at 10%.

4.- Line 192, As I mentioned in point 2, sample size justification, objectives, statistical methodologies, and results must be in line. Reading this Section, it is clear that authors want to demonstrate the superiority of Springfusor compare to standard of care, but the name of the test used to do it, needs to be included, because the sample size depends on the approach.

5.-Line 212, change “The difference was considered statistically significant at p<0.05” by “p<0.05 was set to consider statistical significance” because it refers not only to the difference but also to any hypothesis.

6.- Line 236, Table 1, delete % from the first row.

7.- Line 236, Maybe “Particulars” is not the best denomination

8.- Line 240, Delete “p=003”

9.- Line 248, change “4(CI: 4-5)” and “2(CI:2-2)” by 4(IQR: 4-5)” and “2(IQR:2-2)”

10.- Line 250, change “96% vs 61%” by “96.2% vs 61.4%”

11.- Line 253, change “96.5” by “95.7”

12.- Line 254, Table 3, delete 95% CI in the first row

13.- Line 254, Table 3, change CI for the median pain score by IQR and add CI in the rest of rows

14.- Line 254, Table 3, consider reviewing this sentence “would use it in future she got raised BP in next pregnancy”

15.- Line 255, add IQR=interquartile range

16.- Line 259, Table 4, As I said in my previous review, please use the same name in all tables when referring to the standard (or control) arm and intervention (Springfusor), besides sometime arms are used, in others they are denoted as groups…

17.-Line 259, Table 4, check this information, p-value=0.863 obtained from a Fisher test, this is not possible because this test cannot be used with more than two categories.

18, Line 259, Table 4, check this information “Pain score …n(%)” if you are comparing percentages then Mann Whitney test is not appropriated

19.- Line 260, Chage Fishers by Fisher`s, (if you keep this test)

7. PLOS authors have the option to publish the peer review history of their article (what does this mean?). If published, this will include your full peer review and any attached files.

Reviewer #1: No

Reviewer #4: **Yes: **T. Pérez

---

## [Author Response · Author response to Decision Letter 2]

9 Apr 2024

We have point by point response to the reviewers comments in attachment

---

## [Decision Letter · Decision Letter 3]

29 Apr 2024

PONE-D-23-13036R3Randomized trial to compare acceptability of Magnesium Sulphate administration for preeclampsia and eclampsia: Springfusor pump versus standard of carePLOS ONE

Dear Dr. Ononge,

Thank you for submitting your manuscript to PLOS ONE. After careful consideration, we feel that it has merit but does not fully meet PLOS ONE’s publication criteria as it currently stands. Therefore, we invite you to submit a revised version of the manuscript that addresses the points raised during the review process.

**ACADEMIC EDITOR: Please respond to all reviewers comments**

We look forward to receiving your revised manuscript.

Kind regards,

Ahmed Mohamed Maged, MD

Academic Editor

PLOS ONE

Journal Requirements:

Reviewers' comments:

Reviewer's Responses to Questions

**Comments to the Author**

1. If the authors have adequately addressed your comments raised in a previous round of review and you feel that this manuscript is now acceptable for publication, you may indicate that here to bypass the “Comments to the Author” section, enter your conflict of interest statement in the “Confidential to Editor” section, and submit your "Accept" recommendation.

Reviewer #4: (No Response)

2. Is the manuscript technically sound, and do the data support the conclusions?

Reviewer #4: Yes

3. Has the statistical analysis been performed appropriately and rigorously? 

Reviewer #4: Yes

4. Have the authors made all data underlying the findings in their manuscript fully available?

Reviewer #4: Yes

5. Is the manuscript presented in an intelligible fashion and written in standard English?

Reviewer #4: Yes

6. Review Comments to the Author

Reviewer #4: Thank you for replying to all my comments. I just have two minor comments:

1. Line 197 : Please change this sentence:

To demonstrate a 50% difference in the proportion of women who reported administration of mgso4 was acceptable between those who used Springfusor and standard of care,

by

To detect an absolute increase of 50 percent points in the proportion of women who reported administration of mgso4 was acceptable between those who used Springfusor and those who used standard of care,

2. In Table 5 and Table 6 it is better not to include p-values

7. PLOS authors have the option to publish the peer review history of their article (what does this mean?). If published, this will include your full peer review and any attached files.

Reviewer #4: No

---

## [Author Response · Author response to Decision Letter 3]

1 May 2024

Made the correction as advised and also edited the two tables (5&6) to remove the p values. See the attachment

---

## [Editor Report · Decision Letter 4]

7 May 2024

Randomized trial to compare acceptability of Magnesium Sulphate administration for preeclampsia and eclampsia: Springfusor pump versus standard of care

PONE-D-23-13036R4

Dear Dr. Ononge,

We’re pleased to inform you that your manuscript has been judged scientifically suitable for publication and will be formally accepted for publication once it meets all outstanding technical requirements.

Kind regards,

Ahmed Mohamed Maged, MD

Academic Editor

PLOS ONE
---

## [Editor Report · Acceptance letter]

2 Jun 2024

PONE-D-23-13036R4 

PLOS ONE

Dear Dr. Ononge, 

I'm pleased to inform you that your manuscript has been deemed suitable for publication in PLOS ONE. Congratulations! Your manuscript is now being handed over to our production team.

Kind regards, 

on behalf of

Professor Ahmed Mohamed Maged 

Academic Editor

PLOS ONE